# Blue-Green Smart Mobility Technologies as Readiness for Facing Tomorrow's Urban Shock toward the World as a Better Place for Living (Case Studies: Songdo and Copenhagen)

**Hamid Doost Mohammadian [1],\* and Fatemeh Rezaie [2]**

[1] Department of Business and Economics, University of Applied Sciences (FHM), 33602 Bielefeld, Germany
[2] Management Department, Industrial Management Institute, Tehran, Iran; baharrezaie70@gmail.com
\* Correspondence: Doost@fh-mittlestand.de; Tel.: +49-17620826905

**Abstract:** Nowadays, we are on the cusp of a future that will face many global challenges and crises, as well as unforeseeable shocks of tomorrow. The rapid growth and development of technology will bring forth exponential change that may challenge and threaten our human psychology. Solutions and policies are needed to deal with today's challenges, tomorrow's shocks, and global crises to preserve the world and mankind for the future. In this research, Blue-Green sustainable mobility technologies are introduced as a pathway to create modern sustainable and livable urban areas to tackle these challenges. Clean and inclusive mobility, based on Blue-Green and sustainable infrastructure, low emission greenhouse gases, ubiquitous computing, smartness and digitalization is realized as one of the keys that could make the world a better place for living. This research examines inclusive transportation technology, its indicators and its impacts on creating modern livable urban areas with high a quality of life as a pathway to navigate the cusp of tomorrow. Furthermore, the roles of technology such as Information Technology, Internet of Things, Internet of Business, Internet of Manufacturing, and Internet of Energy as technology tools to develop such mobility is investigated. Literature reviews, surveys, case studies—including Songdo as a ubiquitous city and Copenhagen as a digital and clean city—and revised versions of Kiwi and Kampenhood and BESQoL (built environment sustainability and quality of life) methodologies are the main methods in this study. New concepts of mobility technology and eventuating cultural synergies, as a readiness for facing tomorrow's world crises with a higher quality of life and well-being by using the 5th wave theory, are discussed.

**Keywords:** blue-green mobility technologies; ubiquitous; tomorrow's shocks; sustainability

## 1. Introduction

Nowadays, we have passed the 1st, 2nd 3rd, and 4th industrial waves. Before 1970, fossil energies and various businesses affected and improved technologies and the economy, preceding the growth of Information Technology. Since the 1970s, Information Technology has been able to change and improve various energy and business models, with great impact on our lives; in addition, modern technologies have facilitated the change in global policy from fossil energy and coal to sustainable renewable and clean energies, such as solar, wind, biomass, etc.

These days, we are facing not only global challenges including poverty, environmental challenges, climate change, social instability, insecurity, slums, health diseases and economic challenges, but also future shocks concerning the rapid growth of technologies. Therefore, governments, policy makers, researchers and planners should work hard together to discover solutions and strategies in order to

tackle these challenges, aimed at reducing global concerns such as the emission of greenhouse gases, and promoting economic growth, social inclusion, achieving sustainable development and improving quality of life.

In recent decades, global concerns and controversial phenomena have been investigated in order to find solutions that will reduce these concerns, maintain the world for future generations, and improve livability and the quality of human life. Based on research, sustainable development is one of the best solutions to deal with urbanization challenges and to improve quality of life [1]. New concepts of modern livable urban areas that focus on life satisfaction and sustainability are required to deal with urbanization challenges, in order to create a more livable and sustainable world.

High technologies are tools that can help achieve sustainable development. For instance, rapid and unplanned urbanization has made fundamental challenges in South Korean cities such as Seoul and Segundo. Ubiquitous services founded on technology are chosen as the path to deal with these challenges, and generally, creativity is required to help this country make more livable and sustainable urban areas. Although technology can be applied as a solution to tackle global challenges, it is also a reason for future shocks. In the relationship between technology and global challenges, concepts and solutions such as digitalization, ubiquitous computing and smartness are ideal platforms to face these challenges, although such rapid development can create psychological unease.

In this context, the role of mobility as a component of urban infrastructure in achieving modern, smart, sustainable and livable urban settings is examined. In addition, the role of modern technologies such as Information Technology, Internet of Things, Internet of Business, Internet of Manufacturing, and Internet of Energy in developing Blue-Green Mobility is investigated. According to studies, sustainable transport is one the main tools to develop sustainability as a technique to make modern livable areas [2]. Köhler speculates that sustainable transport is considered a global concept to create a new urban living, to make the world a better place for living [2]. Accordingly, Blue-Green mobility is an environmentally friendly tool to achieve such urban areas, aligned with green strategies, water management policies and the notion of a Blue-Green economy, as a sustainable economy is one of the main parameters for the high quality of life. Fundamentally, Blue-Green strategies and paths would help us in facing such global challenges as pollution, environmental challenges, economic problems, social instability, as well as future shocks. The main aim of the research is to show how Blue-Green mobility could play a role in creating modern urban areas that focus on sustainable development as a tool for making the world a better place for living. In order to approach the main aim, four sub-goals are proposed: (1) the first sub-goal is to find out how Blue-Green mobility could be achieved; (2) the second sub-goal is to indicate different aspects of how Blue-Green mobility could have influence on sustainable urban areas; (3) realizing the importance of creating new smart and sustainable urban areas as tools toward gaining sustainable development is the third goal of the research; (4) the last goal of the research is to show the role of innovation and high technologies such as digitalization, ubiquitous computing, etc., in creating the solutions and pathways toward Blue-Green mobility in order to develop a Blue-Green economy.

## 2. Background

### 2.1. Sustainable Development

Sustainable development is one the most important and controversial phenomena of recent decades [3]. It is concerned with strategies, policies and efforts utilized to improve human well-being and to maintain the world for future generations through the management of human environmental systems [4].

Sustainable development, based on three indicators—social, economic and environmental sustainability—is a solution to provide basic human needs, creating environmental development and protection, achieving equality, ensuring social self-determination and cultural diversity, and preserving ecological integrity in order to improve the quality of human life [5,6].

The authors believe that sustainability has more pillars than the three abovementioned ones. We contend that seven pillars are required to develop sustainability: environment, economic, social, educational, cultural, technical and political aspects form sustainability. These aspects make a puzzle in which all of the segments are directly or indirectly related to each other. Figure 1 presents seven aspects of sustainability (7PS model) and its classification. This model is focusing on seven pillars of sustainability as below:

1. Economy
2. Social
3. Environmental
4. Political
5. Cultural
6. Educational
7. Technical

In order to achieve sustainability all these seven parameters should improve approximately equably. With this model we can calculate the sustainability in each company, business, city, or area. Sustainability has occurred when the figure is more regular. In addition, the ratio could influence the achievement of sustainability. Generally, the blue shape is more sustainable than the red one. Not only is the blue one's ratio higher but also the blue one's regularity is more proper than the red one. These two reasons make the blue shape more sustainable than the red one. In other words, approaching a high ratio of sustainability and developing approximate equability are required to achieve sustainability and sustainable development [7].

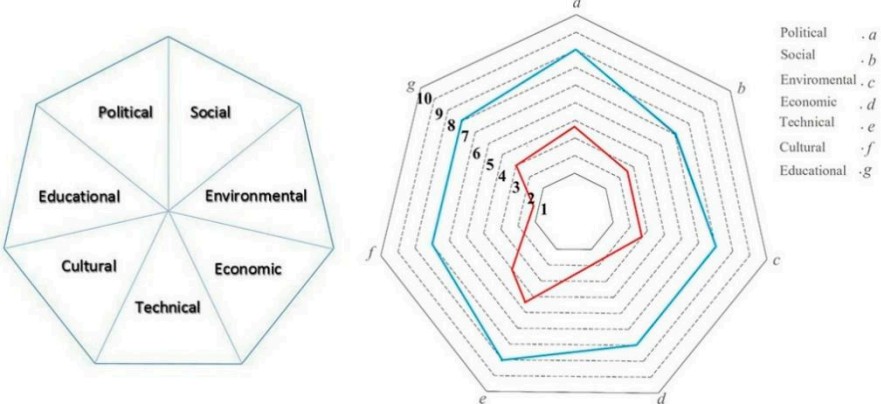

**Figure 1.** 7 PS Model [7,8].

Generally, high quality of livability and life, health and prosperity with social justice, being environmentally friendly and preserving the earth's capacity to support future life are the main aims of sustainable development [9].

*2.2. Mobility*

Mobility is about demand for activities, where the cost is an integral part of demand [10]. Generally, strategies related to mobility try to gain sustainable patterns of consumption and production [11].

Sustainable mobility concerns achieving sustainable people, planet and profit development and the balance among global environment, economic success and quality of life [12]. There are four techniques introduced as ways to gain sustainable mobility:

- Utilizing technology in mobility.
- Land-use development, such as planning and regulations, needs to be integrated, etc.

- Clearly targeted personal information, including social pressure, awareness raising, demonstration, persuasion and individual marketing, is also crucial.
- Regulation and pricing mean that the external costs of transport should be reflected in the actual costs of travel through higher fuel prices or through some form of road user charging [13].

Mobility as a component of infrastructure requires being kept up with the growing needs of rapid urbanization. These days, sustainable and smart mobility is required to make an urban setting livable. Smart mobility could gain through utilizing new technologies, ICT to reduce environmental challenges, provide humans' demands and make livability in cities [14]. Sustainable mobility looks fundamental, although smart mobility looks pragmatic. Fundamentally, smart mobility is the clever and active small brother of sustainable mobility. In other words, sustainable and smart mobility could make mobility not only be kept up with the growing needs of citizens, but also be aligned with sustainable development goals [15,16].

*2.3. Modern Technologies*

We are living in an era in which modern technologies influence various services and eventually every aspect of life. Ubiquitous city, smart city, E-services, Ubiquitous and smart services, IoE-energy management, Industry 4.0 and so on are based on modern technologies like ICT, IoT and IoE [17–19].

● **Information Communication Technology (ICT):**

ICT is about all computer-based advanced technologies to control, manage, communicate and transfer information. Generally, ICT is about digital processing and applying information to manage, store, retrieve and transfer information by electronic computers [20,21].

As information is the centre of social, economic, political and environmental activities, ICT could influence all aspects of life and make significant changes in society. In addition, it is a tool to meet the demands of today's citizens. ICT is a powerful technique that could be utilized in different businesses and services including medicine, healthcare, manufacturing, industry, urban planning, transportation, education and learning, finance, public management, energy production and so on, to improve their productivity, efficiency and processes as well as developing sustainability. The main advantages of ICT are: reducing poverty, improving the education situation through global education, promoting gender equality, empowering women, improving health conditions, enhancing businesses, creating successful businesses with high productivity and efficiency, improving economic sustainability, reducing environmental risks, improving social instability, achieving sustainable development and creating areas based on sustainability and high quality of life [22–24].

● **Internet of Things (IoT):**

Internet of Things is a new version of ICT founded on interrelating embedded systems through sensors and electronic applications. The words "Internet" and "Things" mean an inter-connected world-wide network based on sensors, communication, networking, and information processing technologies [25].

Enabling things to be connected anytime, anywhere; with anything and anyone through any path/network and any service is the main target of IoT that could influence various businesses and services [26]. In particular, the main concept of IoT is based on anything, anyone, any service, any business, any path, any network, any place, any time and any concept.

Although IoT has many advantages like energy consumption, reducing time, money saving, developing smart services, and all of the advantages that ICT has, there are challenges that make the application of IoT difficult. The main challenges in this area are: high cost for specific components, data management, interoperability, technical challenges like proper technology infrastructure, sensors and so on, security challenges, data management, education barriers to educate knowledgeable experts and intellectuals who are aware of applying IoT in different services, etc. Fundamentally, these

challenges are significant threats for IoT usage. It is needed to tackle these challenges to be able to apply IoT to benefit from its privileges [19,27–31].

IoT could transform the way we live today through creating intelligent and smart services, businesses, daily tasks and chores. IoT can make smart and intelligent services like smart building, smart transportation, smart infrastructure, smart businesses, smart life and so on towards designing smart cities that could improve sustainability. Therefore, IoT is a fundamental tool not only to improve environmental and economic sustainability, but also to develop sustainability and create smart and sustainable cities [26,32].

Internet of Things is a new revolution of the Internet that could achieve sustainable development in society and the world in order to improve quality of life in living areas.

- **Internet of Energy (IoE):**

IoE makes changes in sources, loads, developing clean energy, renewable energy, plugging in electric vehicles, etc. [33]. IoE is about utilizing electricity infrastructure, making energy production cleaner and more efficient, causing more power in the hands of the consumer [34]. IoE is founded on dynamic networks connecting the energy network to the internet in order to improve energy efficiency and be more environmentally friendly [35,36]. Improving high technologies such as ICT, smartness, digitalization and so on, influence the application of IoE in businesses to benefit from its advantages [36].

In particular, IoE is a tool to make proper energy management. Energy and the combination of energy with information and communication technology could influence the efficiency and productivity of businesses, even consumers and the World, through its opportunities and privileges [37].

The impact of the IoE on data infrastructure and the selection of a suitable data platform is one of the major challenges of the IoE [38]. Education is a solution to confront with this challenge. Energy efficiency, clean (Blue Green) energy, new resources of energy, being environmentally friendly, cost efficiency and developing sustainability towards better quality of liveability and life are the main benefits of IoE [35,37].

### 2.4. Smartness, Digitalization and Ubiquitous

Smart and smartness have been utilized in different sciences in order to improve function. When new technologies and open systems of innovation started to be manipulated, around the 1970's, the application of smartness in citizens, enterprises and cities occurred to enhance the quality of human life [38,39]. Medina-Borja declared smartness as an idea that covers Internet of Things, artificial intelligence, and all smart services like smart phones, smart cities, etc. [38].

Digitalization, like smartness, plays significant roles in various aspects of life, such as different businesses in recent decades. Digitalization is about the integration of digital technologies into everyday life, which can be digitalized. It could be applied as one of the tools to develop sustainability in countries as well as in businesses [37,40,41].

Enhancement of high technologies like ICT, IoT, IoE and so on, create and develop new concepts such as ubiquitous and smartness. In addition, robotics, machine learning and virtual reality influence the evaluation of ubiquitous and smartness [42].

Ubiquitous is derived from the noun **ubiquity**, meaning omnipresent or being present everywhere or in many places at the same time. The term ubiquitous is concerned with overstating things and people that seem to turn up everywhere and has been more popular than ubiquity [43].

These days, ubiquitous computing is utilized in several aspects of life including urban development and planning, banking, shopping, learning, energy management and so on, which would influence humans' life and sustainability [43–45].

### 2.5. Industry 4.0

As it was mentioned, in the second half of the 20th century through developing industrialization, industry 4 has emerged [46]. I4.0 was introduced by the German government, founded on technological

changes in manufacturing and policy frameworks for companies in order to survive in global competition [47].

I4.0 is about intelligent and smart networking of products and processes based on five technology areas: Embedded systems, smart factories, strong networks, cloud computing and Information Technology (IT) security [48]. I4.0 plays significant roles in the success of organizations. The main benefits of I4.0 are:

- Reducing cost including production costs, logistic costs and quality management costs
- Creating more friendly and effective environment
- Establishing sustainable and effective energy management
- Improvement of mass production
- Improving customer services and products
- Reducing processes of releasing new product to market [49]

The main challenge of I4.0 is concerned with adjusting leaders, managers, entrepreneurs, labour and generally capital to utilize new methods and processes based on I4.0 and creating significant changes. To deal with this challenge, education, and training play vital roles [41,50]. In particular, I4.0 is a technology that is able to improve businesses performances, processes, their value chain and services to develop successful and sustainable business with high efficiency and productivity.

*2.6. Society 5.0*

The rapid evaluation and improving of information and communications technology (ICT) and other kinds of high technologies like digitalization make significant changes in industry and society.

The Society 5.0 concept is one of the ideas created through these evaluations by Japan. Basically, society 1.0 is hunting society emerging upon the birth of human beings. At 13,000 BC through the development of irrigation techniques, society 2.0 named as Agrarian society was started. Industrial society as society 3.0 founded on the invention of the steam power machine and mass production emerged at the end of 18th century. Society 4.0 is an information society via improving technology at the second half of the 20th century, and society 5.0 as super smart society has emerged from the 21st century [51].

Society 5 is concerned with every aspect of life such as mobility, manufacturing, food production, reducing disasters, energy, finance, public services, cyberspace, and efficiency of organizations, regions and cities [17,19,52]. Fundamentally, society 5.0 is a tool to make a balance between economic development and social sustainability. Figure 2 declares economic advantages and social enhancement of society 5.0 [53]:

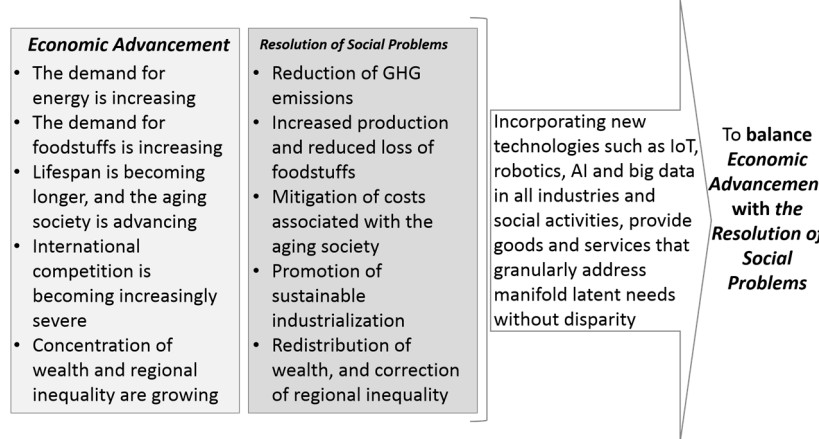

**Figure 2.** Society 5.0 Advantages [53].

## 2.7. Hybrid Companies and SMEs

A new era of businesses like hybrid SMEs could have an influence on developing sustainable and smart mobility. Such businesses are based on using internet of things to produce better services and products as well as expanding their markets. Hybrid businesses would achieve their targets, profit, efficiency, and social beneficiary goals through applying technology [17].

Hybrid SMEs have two main aspects:

- Business Economy focused on Financial
- CSR Strategies concerned with social responsibility, being environmentally friendly, energy and resource saving, future planning and so on.

## 2.8. Blue-Green Infrastructure

Growing urbanization increases global concerns which threaten the future of the world and human well-being. In these days, green strategies are not sufficient enough to face with these challenges and able to be environmentally friendly and aligned with sustainable development [30,54,55]. Besides, of green strategies, blue strategies concerned with stormwater management, water security, flood management, tree health, and recreation needs are required to tackle global challenges. Therefore, Blue-Green Infrastructure is a one of the ways to tackle global challenges and create livable settings through restoring the water cycle back to a nature balance while improving the city's amenities by effectively managing water and promoting green infrastructure. Blue-Green infrastructure has two parts: landscape planning concerned with open space strategy, urban forest strategy, biodiversity strategy and environmental strategy and water planning including sustainable water use plan, integrated water management strategy, flood studies and stormwater management plan [54]. Generally, the main targets of Blue-Green infrastructure are to find out opportunities for integrated greening and water management outcomes, to create a framework for collaboration among council departments, key industry and community groups to design a new blueprint and concept for living areas [55].

## 2.9. Livability and Quality of Life

Quality of life and livability are indicators that could influence the satisfaction of humans about their lives as well as sustainability.

Quality of life is a multidimensional concept including physical, socio-cultural and psychological dimensions as well as environmental ones. Different multiple life domains like housing conditions, education, employment, work-life balance, access to institutions and public services, and their interplay; would create quality of life indicators and eventually influence humans' wellbeing [55]. Fundamentally, the quality of urban life is determined by physical aspects such as the quality of the built environment and its facilities, even social indicators and human ties in the community. So, there is a relation among sustainable development and improving quality of life [56,57].

Liveability as well as Quality of life could influence human satisfaction with their life. Livability is about the more objective indicators of economic wellbeing, including human capital, to the more subjective quality of life indicators such as social capital, qualitative expressions of personal satisfaction and so on. Generally, livability is applied as an indicator to describe the overall contribution of the urban environment towards enhancing the quality of life or wellbeing.

So, mobility and transport systems, affordable housing, education etc., could influence livability. In particular, sustainable development is required to achieve a high quality of livability.

## 2.10. The 5th Wave/Tomorrow Age Theory

The world has been changing through human civilization development and improving technologies every day. Based on industrial revolution changing from mechanization by water and steam power, mass production to information technology; four waves were realized. The 5th theory is about the proceeding of the future of Industry 4.0 and Society 5.0 (I5.0 and Society 6.0),

and edge of tomorrow has been invented by Prof. Dr. Hamid Doost Mohammadian for the first time in 2010 and has been improved between 2017 and 2019.

The world has been changing through human civilization development and improving technologies every day. Based on industrial revolution changing from mechanization through water and steam power, mass production to information technology; four waves were realized: the first age named as the agriculture age, the second wave introduced as the industrial age, the third wave named as the post-industrial age and the fourth wave introduced as the digitalization age [31,41].

*First Wave (Agriculture Age):*

The first wave, introduced as pre-industry period or industry 0.0, commenced around 70,000 years ago through fire, light and wheels. This focused on mechanical production and agriculture industry.

*Second Wave (Industrial Age):*

The second wave happened around the 17th century when steam power, mechanization, chemical industry and water machines named industry 1.0, besides mass production, assembly line and electrical energy being introduced as industry 2.0 was applied.

*Third Wave (Post Industrial Age):*

The advent of the Digital Revolution in the 20th century, created industry 3.0 based on developing of computers, automation, electronics, information and communication technology.

*Fourth Wave (Digitalization Age):*

The fourth wave is introduced also as I4.0 has emerged in the second half of the 20th century through digitalization and automatization of every part and manufacturing process of company. It makes not only huge changes in production, but also in every aspect of life [31].

The 5th wave/tomorrow age/final age theory is about the proceeding of future of I4.0 and Society 5.0 and edge of tomorrow with a focus on today's challenges and tomorrow's shocks. Based on this theory, modern SMEs that are not only concerned with technological business with financial and economic approaches, but also social responsibility like CSR strategies, being environmentally friendly, improving quality of livability and life.

Generally, the 5th wave theory is a tool to achieve Blue-Green sustainability introduced by authors and required to deal with urbanization challenges as well as preserve the world for the future. Blue-Green sustainability is a kind of sustainability founded on seven pillars including economic, environmental, social, cultural, technical, education and political sustainability that is environmentally friendly, even aligned with green strategies and water management [31].

The 5th theory is about how to be ready for the edge of tomorrow due to today's challenges and tomorrow's shocks based on technological consequences. In particular, based on 7PS model, seven pillars of sustainability such as environmental, economic, social, cultural, technical, political and educational sustainability play significant roles in this theory and help us for tomorrow readiness [31].

This theory and its dimensions could make modern SMEs aligned with successful business economy, CSR and sustainability strategies like smart technology SME management, high technologies and HR competencies. Such SMEs are able to improve livability and quality of life indicators as well as economic sustainability and blue-green mobility. Therefore, the 5th wave theory is a tool to create modern technologies and solutions that could confront with future concerns via HR competencies, implementing, developing and applying high technologies like high technologies for blue-green mobility technologies. High technology influences the achievement of such SMEs in the sustainable mobility sector. Technological development has led to new opportunities for business improvement. Internet of Technology (IoT), Internet of Business (IoB), Internet of Energy (IoE), Internet of Manufacturing (IoM), cyber-physical systems, big data, AI and machine learning are new techniques used in the I4.0 that enable SMEs to better manage resources and increase flexibility to respond to the business conditions.

Fundamentally, the 5th wave theory could be introduced as a technique to get ready for the edge of tomorrow, today's challenges, and tomorrow's shocks in the field of transportation to make the world a better place for living.

The world confronts global challenges such as poverty, slums, insecurity, social instability, economic problems, energy and environmental concerns that threaten the future of the world and humanity. It is vital to find out solutions to tackle these challenges in order to maintain the environment and the world for future generations and improve livability of areas at present.

Creating successful and sustainable modern hybrid SMEs are a kind of solution to deal with global challenges. In other words, sustainable and successful modern hybrid SMEs based on utilizing high technologies, HR competencies for urban planning and mobility, proper management like smart energy management, IoE and CSR strategies could achieve sustainable development as well as successful marketing towards economic sustainability. Therefore, these SMEs based on the energy sector could be used as a solution for a good economic situation and sustainable development.

According to the latest research, many devices and equipment are needed to implement technologies like IoE in businesses. What is the reason? As reports mentioned, poor growth of IoT technology in companies is due to the lack of sufficient knowledge and skills in the HR to apply the relevant techniques. Identifying and defining the competencies needed to utilize this blue-green mobility technology and incorporate it into job competencies along with proper planning for human resource development is one of the essentials of today's business.

Generally, blue-green mobility based on IoT technologies, IoE and smart energy management could be applied as tools and motivation towards improving HR competencies in order to develop modern hybrid SMEs like SME 4.0. Such SMEs are able to participate, compete and survive in markets as well as cultivating sustainable development. The 5th wave theory founded on HR education/training and utilizing high technology in businesses is a path to create modern innovative SMEs concerned with CSR strategies, environmental sustainability and sustainable development as well as successful business that are able to conquer future concerns towards a more sustainable and livable World (balancing between CSR strategies and business technologies).

## 3. Methodology

To achieve the aims of the research, the main question should be answered.

How would Blue-Green mobility based on high technologies make the world a better place for living?

In order to find out a logical answer to the main question and aims of the study, sub questions are proposed:

1. What is Blue-Green and inclusive mobility technologies?
2. How would sustainable mobility influence the livability of urban areas and achieve sustainable development?
3. Which dimensions are required to achieve Blue-Green mobility?
4. What is the role of high technologies like ICT, IoT, IoB, IoM and IoE on developing Blue-Green mobility technologies?
5. Why is it vital to alter urban settings into modern, sustainable and livable areas?
6. How to use the 5th wave theory to prepare for facing tomorrow's shocks?
7. How Blue-Green mobility technologies can help us as readiness for facing tomorrow's shocks to make the world a better place for living?

To find out logical answers for the research questions and attain the aims of this study, it is important to apply proper methodology. In this paper, three stages were developed to implement the study:

**Stage 1: The development of a research methodology**

This stage was based on the literature review with the focus on the area related to the objectives of this research. In particular, ISI articles since 2008 were applied for analyzing and indicating results,

debates and discussion of this research. The authors have explored papers published in the context of sustainability, sustainable development, smart and sustainable mobility, livability and quality of life. Furthermore, the importance of sustainable and livable urban areas based on Blue-Green infrastructure and economy was discussed.

**Stage 2: The development of the research tool**

The research tool was developed on the basis of the relevant literature, knowledge, questionnaire and personal experience.

The purpose of the surveys was to indicate policies, strategies, and solutions in order to find out the roles of infrastructure and comprehensive strategic urban planning in creating sustainable and livable urban settings. Impacts of proper mobility as a component of urban infrastructure in altering the world as better place for living through creating sustainable and livable areas were investigated. In addition, roles of technology in achieving Blue-Green and inclusive mobility are realized. The methodology of the revised Kiwi and Kampenhood and BESQol (built environment sustainability and quality of life) were used as a tool for analyzing.

Kiwi and Kampenhood methodology is used in social science, engineering, and management, and comes from the Persian Zarathustra (good thoughts, good words, and good deeds). In 2009, Hamid Doost Mohammadian used dialectical thinking related to this methodology and designed a diagram to revise the method. The aim of this methodology is to find out the factors influencing people's well-being. This methodology presented in Figure 3; is based on three levels with seven questions and analysis had been done through these questions [57].

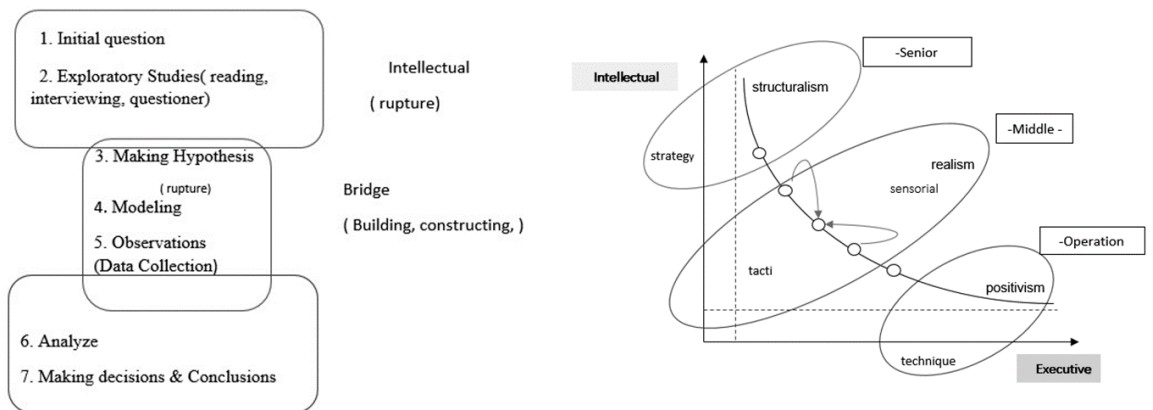

**Figure 3.** Revised diagram concerning the Kiwi and Kampenhood methodology and its levels [7].

BESQol is another methodology, which is associated with the built environment and designed for developing sustainable low carbon that provides capabilities for a high livability and life quality for all members of the community. This methodology includes examining five categories relevant to the development site: the natural environment and natural capital, the built environment, movement, economics, and human capital and quality of life [18,58].

**Stage 3: Gathering the study results and analyzing**

Through literature reviews, case studies, surveys and analyses, some results were obtained; these results were analyzed and discussed by the authors in the discussion section.

In particular, ISI articles published since 2008 were used for analyzing and finding out results in this paper. Based on the lack of theories and analyses related to this issue and ubiquitous field, case study is a proper tool and method for exploring and discovering in this field. In addition, a questionnaire was applied to analyze more precise results.

According to the quality of life assessment system (presented in Figure 3), a questionnaire among 50 experts in the sustainability field done and, via Kiwi and Kampenhood methodology, analyses were

considered. In addition, the instrument used for the purpose of data collection in the quantitative and qualitative phase of the study was the Professional Identity Questionnaire being adapted from Kao and Lin (2015). This questionnaire consists of 22, 5-point Likert scale items ranging from 1 (strongly disagree) to 5 (strongly agree). Different questions were proposed to measure relations among sustainable and smart mobility technologies to indicators concerned with quality of life and livability including environmental concerns, health, social stability, security, political, country development and country infrastructure, water management, education and technology.

## 4. Case Studies

### 4.1. South Korea/Songdo

South Korea as a pioneer of designing and creating U-city confronts challenges related to rapid urbanization, supplying citizens' needs, environmental challenges, contamination, etc., applies ubiquitous services in different cities to struggle with these challenges and making better urban areas.

U-City emerged in the political arena first in 2004 and then as the second step toward the realization of the ubiquitous society, the enactment of the 'Act on Ubiquitous City Construction' has occurred since 2008 in Korea. Physical, spatial urban development with ubiquitous technologies are combined in the U-city. So, the limitation on physical distance and time could be overcome and a new urban model for a sustainable, intelligent city will be developed [59].

For the first time in 2007, the U-Eco City founded on high technologies, ubiquitous and ecology was developed in South Korea. The project lasted between 2008 and 2013. The U-Eco City is declared as a "sustainable future green city made innovative city based on integration of ubiquitous and ecology technology into city space. The main characters of this project are:

- Eco city planning, design, evaluation based on u-technology
- Construction technology of the u-water circulation system
- Construction technology based on low-energy and recycling
- Eco city construction fusion technology based on u-technology

The main goal of this project is to design a future oriented, sustainable city where city management technologies based on ubiquitous infrastructure and the ecological system are combined and would create a comfortable urban environment for urban citizen" [60,61].

Songdo, located on an island about 40 miles from Seoul, is another U-city in South Korea. The high population density in South Korea and environmental problems such as high carbon emission, low sustainability, etc. are the main reasons of creating the U-city (www.casaasia.eu). Songdo is a ubiquitous and smart city built from scratch. This city is "planned in social and physically 'virgin land' (that is, with no former residents, buildings or infrastructures), in policy-protected arenas (with loose, experimentation-oriented and flexible regulations), aiming to attract new residents and companies to be simultaneously users and developers of new smart-IT solutions".

Songdo is realized as a self-ubiquitous (eco city or U-eco city). The main aim of this city is to improve Korea to be more sustainable and greener. In addition, it is nominated as the principle business hub in Asia. Its strategic location, advanced infrastructure, and alignment with a business-friendly environment make this city as a core of business in Asia [59]. Besides being self-ubiquitous, innovation and technologies used in the urban system and infrastructure make Songdo an important sample of U-cities. The main aim of the city is to become an international destination known for the high quality of life provided to its inhabitants and sustained by a vital public realm rich in cultural and recreational attractions. Improving the global economic environment, energy efficiency, the future viability and health of the city, and reducing environmental challenges and climate change are the sub goals of constructing Songdo.

The master plan of Songdo is certified under the Leadership in Energy and Environmental Design (LEED). There are six categories proposed for achieving the main goals related to infrastructure:

- Open and green space: access to nature, sunlight, healthful recreation, public gathering spaces
- Transportation: multi-modal transportation including walking and biking, clean energy; sustainable and smart transportation
- Water consumption, storage, and reuse: reduced water use, stormwater and grey-water recycling, green roofs to reduce runoff, mitigate heat island effect and provide native species habitat
- Carbon emissions and energy use: ASHRAE standards, co-generation, solar energy generation, renewable energy, reduced energy use, pneumatic waste collection
- Material flows and recycling: construction waste recycling, local materials
- Sustainable city operation

Generally, the U-Life services such as U-culture and U-life, U-education, U-business and finance, U-environment, U-health, U-transportation, and U-government provide the digital infrastructure of Songdo. It is said that: "U-life envisions establishment of a resident-friendly community that thrives in a sustainable green living environment with exploitation of the state-of-the-art technologies to the fullest. Building a truly ubiquitous community designed to bring ubiquitous culture, society, and lifestyle to each resident is our vision".

Fundamentally, Songdo may become a positive model for sustainable urbanism in the age of ubiquitous global and cyber culture [59–61].

Fundamentally, every aspect of public life in Songdo founded on technology, ranging from an integrated public transport system to the government's emergency warning system, make Songdo an ubiquitous city. Ubiquitous services not only could deal with Seoul's challenges, but also could develop sustainability and improve citizens' lives. In particular, ubiquitous services make Songdo a liveable and sustainable urban setting, even reducing global challenges that it faces. Therefore, U-services and U-city are innovative solutions for cities like Songdo or Seoul that face global challenges and rapid urbanization to tackle challenges and create sustainable and liveable urban areas.

### 4.2. Denmark/Copenhagen

Smart cities are a solution to be aligned with humans' needs as well as developing sustainability. Therefore, many European countries choose this solution as a path to deal with challenges they face with like environmental concerns, climate change, social instability, economic challenges and so on, even to be kept up with rapidly changing humans' needs.

Smart city was introduced and coined in 1990s. Smart cities are about an idea based on implementation of user-friendly information and communication technologies developed by major industries for urban spaces. Such cities are able to deal with global challenges, climate change, urbanization and scarce resources. It means that smart cities could achieve sustainable development towards high quality of livability and life. Various indicators like smart mobility, smart economy, smart governance, smart society, smart and sustainable energy management, and different kinds of smart services are required to develop smart cities., ICT, IoT, IoE and digitalization play significant roles in developing smart services. In other words, technology is a fundamental component of smart cities [61–63].

Based on Copenhagen Cleantech Cluster research, three main indicators were presented for infrastructure of Smart Cities:

- Physical, including roads, energy grid, bike lanes, district heating and cooling, sewage system
- Communicative, containing standardized coding language, open interfaces, open source technology, citizen inclusion through ICT
- Digital services and systems concerned with broadband, cloud computing, fiber optic cables, sensors, smart phones, mobile networks, databases and so on [62,64].

The OECD introduces Denmark as the most energy secure and sustainable country. The country tries to reduce dependency on foreign sources of energy to zero and become self-sufficient in its own energy production and use. [62].

Different strategies are applied by Denmark to be able to enhance smartness in Copenhagen and other cities. Smart mobility and transportation are realized as the main policies used by the Danish government towards smartness. Smart transportation systems are the efficient and sustainable transportation of people and goods through the utilizing of intelligent technologies such as:

- Independency on fossil fuels via applying clean and inclusive energies such as biofuels, etc.
- Clean transportation systems based on clean and renewable energy
- Bike as main transportation system
- Intelligent transportations systems (ITS) including signal and fleet management
- Improvement of public transportation systems
- Proper city infrastructure based on smartness
- High standards
- Reducing consumption energy by sustainable and smart construction and architecture
- Improvement of cultural norms and education systems
- Appling facilities such as proper insurance, high education, employment and so on, towards high standard of living
- Applying new processes and technologies in industry, transport, and others to make higher efficiency and productivity with lower consumption of raw material and energy
- Converting risks to new opportunities by innovation, high technologies, and digitalization
- Applying sustainable and smart strategies and policies in mobility management, city logistics etc. [64,65]

Not only government policies and strategies, but also smart services and sustainable strategies and solutions used by all Danish mobility organizations improve sustainability of the country. For instance, Maersk is a qualified example. Maersk is a Danish business conglomerate and the largest shipping company, which works in different sectors including transport, logistics and energy sectors. It pursues and applies policies, strategies and paths to develop Blue-Green, smart and sustainable mobility such as reducing $CO_2$ emissions by 40%, plying bio fuels to the owners of its fleet, set to be carbon neutral as main goal in 2015 and using special category management including: stakeholder management, category strategy and sourcing, supplier relationship management, proactive category management and renegotiate agreements and special risk management in different categories. All these policies make sustainable and smart water transport aligned with being environmentally friendly.

In particular, the idea that Copenhagen can become a smart city has developed from the ambitious vision of becoming the world's first carbon-neutral capital by 2025. Copenhagen could be smarter by two strategies:

- Copenhagen is to be the world's leading testbed for smart and sustainable solutions. By turning into a living lab for new green solutions, the city is able to attract innovative companies.
- Copenhagen is very much a believer in 'sharing is caring', as the successes and knowledge gained in the city are to be shared with other cities around the world, and vice versa [65].

Fundamentally, strategies and policies used by the Danish government as well as those smart citizens make Copenhagen and other Danish cities smarter. Smart cities are able to reduce global challenges, supplying growing human needs to create areas with high sustainability, quality of livability and life.

## 5. Results

Based on research, studies, exploring case studies, analyses of articles [7,8,10,11,13–15,19,20,30,31, 35–37,40,41,43,46,57,61,66,67], the following was concluded:

Global challenges including poverty, slums, social instability, health diseases, insecurity, economic problems, climate change, environmental challenges; and future shocks are significant threats for

livability of the world and humanity. Sustainable development and smart citizens could be solutions to tackle these challenges. The new concept of cities founded on achieving sustainable development and training smart citizens could make the world a better place for living.

In addition, a questionnaire was done to find out how smart and sustainable mobility technologies would influence each quality of life indicator more. In other words, different intensities among such mobility and quality of life indicators existed through the questioner results.

Table 1 presents the most intense connection among smart and sustainable mobility technologies and quality of life assessment system indicators.

**Table 1.** Questionnaire Result.

| Quality of Life Indicators | Score |
| --- | --- |
| Environment Sustainability | 4.5 |
| City Infrastructure and Development | 4.6 |
| Economic Condition | 4.1 |
| Security | 3.7 |
| Health and Diseases | 4 |

Based on questions, it was realized that sustainable and smart mobility technologies would definitely improve environment sustainability and then it could improve city infrastructure and city development. In addition, security, social stability, and health could be enhanced directly by applying smart and sustainable mobility technologies. Generally, other indicators related to quality of life like political sustainability, water management and so on, could be enhanced by such mobility, however, at the first stage there is a fundamental relation among these five indicators and smart-sustainable mobility technologies.

In particular, smart and sustainable mobility could reduce environmental concerns such as air pollution, noise pollution, water contamination, climate change and so on. Based on results, such mobility could be a tool for environmental sustainability. Fundamentally, mobility as a component of city infrastructure could directly and indirectly influence on city infrastructure, and the results could show that. In addition, sustainable and smart mobility is a tool or indicator for city development towards modern and smart cities. Economic sustainability is the other factor that mobility systems influenced. There is a relation among mobility and security as one of main quality of life indicators. Security is a social parameter that could be improved by sustainable and smart mobility systems and the results showed this fact. It was determined as one of the five main indicators. Furthermore, sustainable mobility would reduce disease and improve health of citizens. Although, mobility and transportation systems influence the other quality of life indicators like social sustainability, technical sustainability, cultural norms and so on; the relation among these factors and mobility technologies are less regarding the results that aren't presented in Table 1.

Fundamentally, mobility as a component of urban planning could influence city infrastructure and city planning as well as environmental sustainability. Smart, sustainable, and ubiquitous mobility based on water management and environmentally friendly approaches, introduced as Blue-Green mobility by authors, could be a tool and path to introduce new concepts of urban settings that are more livable and sustainable. Technology is a fundamental tool to make such kinds of mobility towards smart, sustainable, and livable urban areas. Such cities are able to deal with global challenges and develop sustainability towards better quality of livability and life.

## 6. Discussion

These days, the world confronts global challenges as well as future shocks, which threaten human beings and the future of the world for living. So, it is vital to deal with them to maintain the nature and humanity. Figure 4 the 5th wave/tomorrow age theory, illustrates the ages, and technologies that we had passed and the point that we are on:

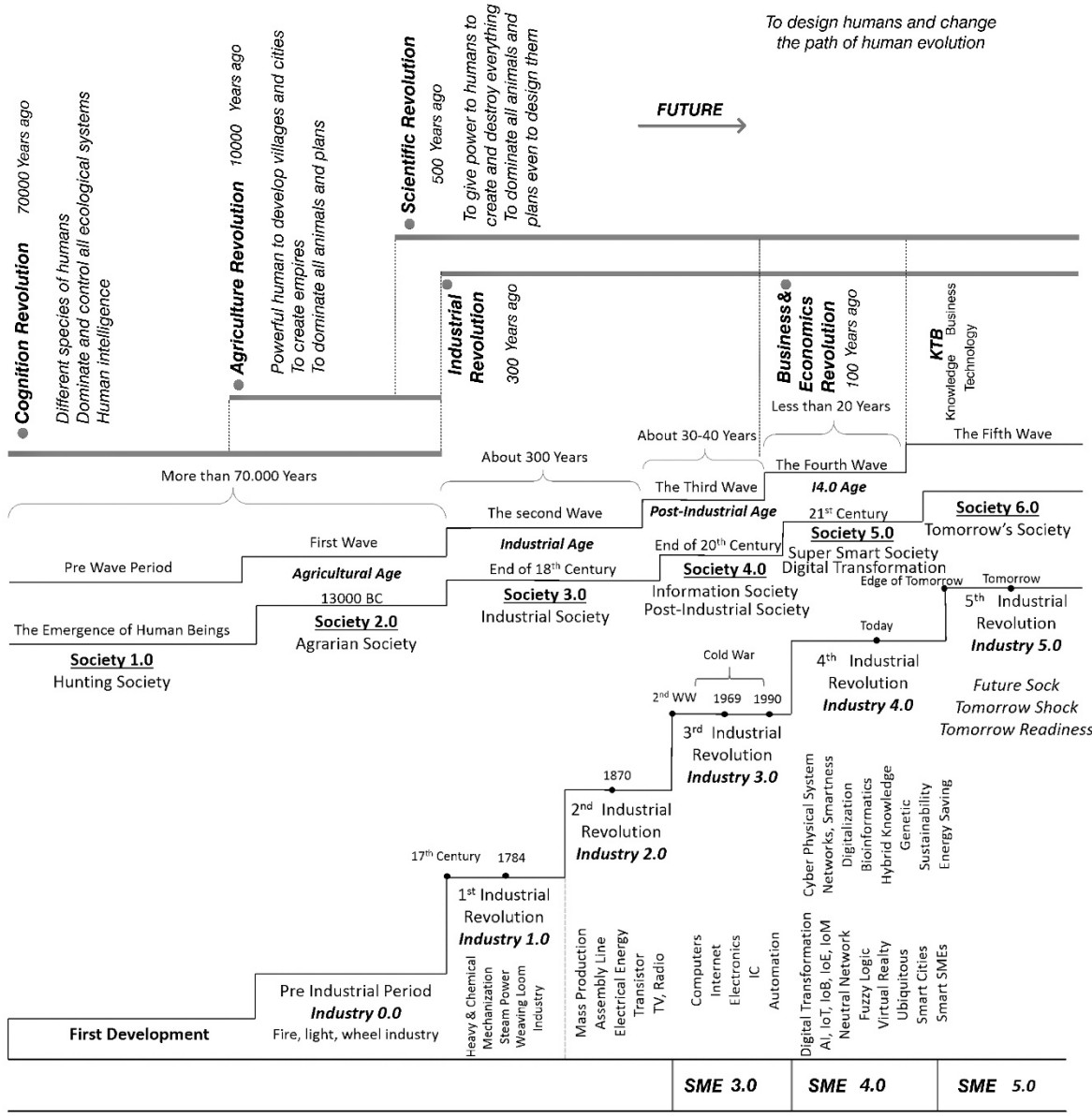

**Figure 4.** The 5th wave Theory, Ages and Technologies [8].

It is required to deal with global challenges to improve quality of life and livability at present, maintain the world for future and be ready for future shocks. In particular, urban development is vital in global transformation to achieve sustainability in order to tackle global challenges, confront with future shocks, and eventually make the world more livable and sustainable. Creating modern, sustainable and livable urban settings is a kind of key to achieve these aims. To design such areas, proper comprehensive urban planning concerned with improving sustainability, being environmentally friendly and being aligned with humans' needs is required. Ubiquitous cities in the East and Digital cities in the West are based on appropriate and comprehensive urban planning. These concepts are solutions to make the countries more livable and sustainable through dealing with global challenges, supplying citizens' needs and improving indicators needed for high quality of life. Basically, these two concepts are based on utilizing high technologies like ICT, IoT, IoE and so on. Therefore, technology plays significant roles in evaluating ubiquitous and digital services. In other words, technology is a tool to invent and manipulate modern strategies, policies, and solutions to tackle global challenges and improve quality of life. Although technology is vital, it is also be a threat. In particular, technology is also a factor for future shock. It is required to train smart citizens as well as applying technology

in different services for better life. Smart citizens who are aware of technology and modern services like ubiquitous services and digital services could have an influence on the confronting of future shocks [18,19,30,41,61,66].

In this research, Blue-Green concept is recommended by authors as a concept to design comprehensive urban plans. Blue-Green strategies are applicable policies that made Blue-Green cities aligned with not only environmentally friendly, but also water management approaches. Nowadays, green strategies are not sufficient to develop sustainability and focusing on both of these phenomena are required to maintain the world and improve sustainability. Environmental concerns like emission of greenhouse gases, climate change and environmental risks; water, wastewater and water management are the main crises for preserving the world and humanity. So, Blue-Green and sustainable urban areas based on Blue-Green and sustainable comprehensive urban plan are proposed as a path to change the urban settings. Such urban areas could make high quality of livability and quality of life through dealing with environmental challenges and improving sustainable development.

Technologies play a significant role in progressing Blue-Green strategies not only in mobility services, but also in all services. Blue-Green and modern city like Ubiquitous and digital city, are founded on technology, sustainability, and innovation. In other words, ubiquitous, digitalization and smart concepts all are based on modern technologies like virtual reality, machine learning, artificial intelligence, robotics, ICT, IoT, IoE and so on. Therefore, it could be declared that technology is a tool to create ideas that are required for modern lives.

Fundamentally, enhancement of technology is a key to be able to create solutions, strategies, and paths to deal with global challenges and be aware of future shocks. In this study, the role of technology in developing Blue-Green and inclusive mobility is realized.

### 6.1. Blue-Green Mobility Technologies

Mobility as a component of urban planning, could influence the quality of livability and life. In other words, the world's natural environment, social well-being, economic development, and livability depends on transportation systems. Transportation has four main categories including (1) *Road Transport*, (2) *Air Transport*, (3) *Water Transport* and (4) *Rail transport*. Each of these transportation systems has specific risks and concerns that could be threats for sustainability. For instance, air contamination of rail transport, water pollution of water transportation, accidents related to rail and road transport, contamination made by air transport, disturbing nature to design roads and rails for transportation, insecurity of public transport, etc., are the main mobility risks that influence social well-being and sustainable development. Therefore, Blue-Green mobility as a sustainable and proper kind of mobility is declared as a path to design Blue-Green infrastructure by the authors.

Blue-Green mobility is a kind of mobility based on green and blue strategies. Green strategies focus on policies like sustainability and being environmentally friendly, sustainable and smart water management policies. Green strategies are used to deal with sustainability and environmental challenges. Sustainability has three traditional pillars including environmental, social and economic sustainability. However, the authors believe that sustainability has seven pillars: economic, social, cultural, technical, educational, political, and environmental. So, green strategies help countries to improve these seven pillars of sustainability through mobility systems. Blue mobility related to water management could influence water-based concerns such as water pollution, water shortage, water refining, etc. In recent decades, sustainable and green strategies are not sufficient to maintain the planet and humanity. Besides them, strategies concerned with water management are required to struggle with urbanization problems. Therefore, Blue-Green mobility could create sustainable and livable urban areas to struggle with urbanization problems through being aligned with environmentally friendly and water management approaches. In addition, Blue-Green mobility is a path to be kept up with humans' demands in order to supply humans' needs. Therefore, Blue-Green mobility is an inclusive mobility concerned with environmental and water management risks in order to reduce them [35–37,39].

Inclusive mobility improves social responsibility in a city. In addition, environmentally friendly strategies make mobility aligned with the environment and avoid destroying it. Mobility based on clean energy such as hybrid vehicles is a good example of environmentally friendly mobility. For instance, high environmental and technical standards of mobility reduce water pollution of water transport; if rainwater management in a city isn't proper, flash floods could occur.

Fundamentally, Blue-Green mobility could create the Blue-Green economy that is needed for high livability and quality of life. Based on the authors' opinion, sustainable economy could improve other pillars of sustainability in order to improve seven pillars of sustainability towards sustainable development. Figure 5 shows how Blue-Green mobility technologies improve Blue-Green economy towards sustainable development:

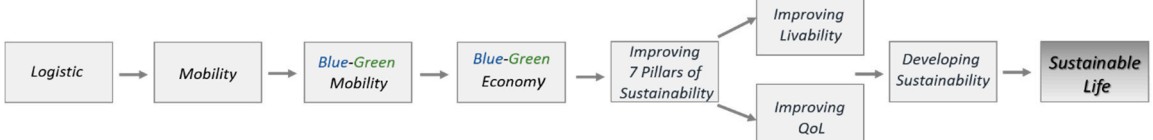

**Figure 5.** Blue-Green Mobility Technologies.

The main privileges of Blue-Green mobility are:

- Social Equity through improving safety and security, privacy, improving health and wellbeing and so on.
- Transport system based on clean energies.
- Better and more efficient traffic management.
- Access to services anywhere and anytime for everyone.
- Safer, greener and cleaner mobility.
- More sustainable and smart transportation.
- Further clean logistics.
- Achieving sustainable and smart urban planning.
- Reducing environmental challenges through utilizing clean energy, decreasing green gas emissions, noise pollution, air contamination and so on.
- Improving economic sustainability through promoting productivity, attracting investment and creating jobs, improving export transportation and so on.
- Being aligned with rapid growing of citizens demands.
- Impact on creating smart, sustainable and ubiquitous infrastructure towards smart and ubiquitous city.
- Enhancing sustainability in urban areas.
- Making quality of life and livability higher in urban areas.
- Making the world more sustainable and livable through reducing global challenges especially environmental concerns.

Fundamentally, mobility as a component of urban planning could influence the infrastructure of urban areas. Figure 6 presents how mobility technologies could achieve Blue-Green infrastructure and the main benefits of such infrastructure:

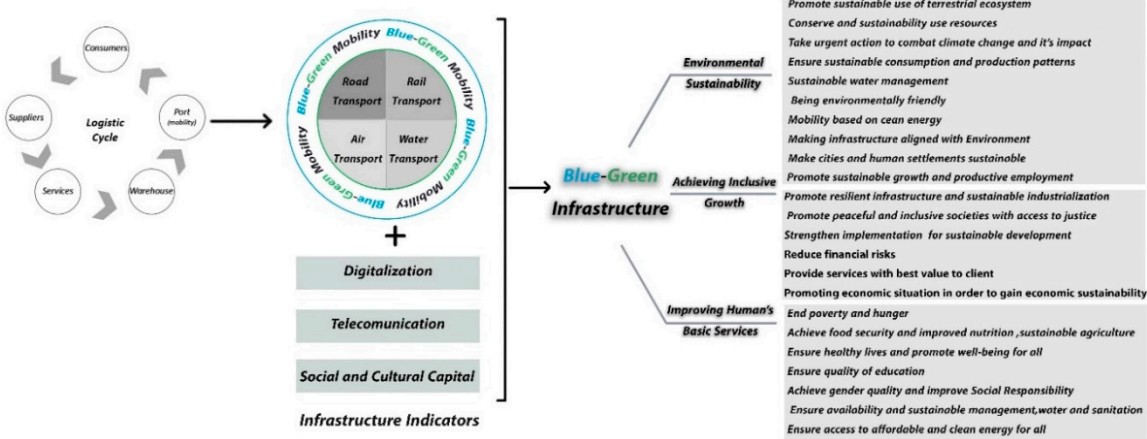

**Figure 6.** Mobility-BGI Theory.

Blue-Green mobility is aligned with environmentally friendly and water management policies. It is able to reduce air contamination, water pollution and eventually all environmental risks. Furthermore, such mobility via environmental sustainability could achieve other aspects of sustainability especially economic and social sustainability. Using clean energies, applying sustainable risk management, creating hybrid and sustainable businesses related to mobility and so on, are solutions to design Blue-Green mobility. In addition, bridge construction, rail construction and urban construction could influence Blue-Green mobility. Generally, mobility is a component of urban planning; so, it could make proper and sustainable infrastructure towards Blue-Green cities.

*6.2. High Technologies and Blue-Green Mobility*

Blue-Green and inclusive mobility need different tools like technology, innovation, proper strategies and governance, smart citizens and so on to be developed. Basically, technology plays important roles in creating clean and inclusive mobility. Solutions and strategies applied in mobility and other services of ubiquitous cities, digital cities and smart cities are all based on technology. For instance, Ubiquitous transportation (U-transportation) founded on ubiquitous technology and computing, smartness and intelligence is the main reason for this enhancement.

One of main indicators of Blue-Green mobility as a clean and inclusive mobility is technology. Like ubiquitous and smart transportation, technology plays a significant role in making Blue-Green mobility. Industry 4.0 is a fundamental concept to achieve smart, intelligent and sustainable mobility towards Blue-Green mobility. In other words, IoT and different kinds of IoT like IoE as components of I4.0, are needed to create this kind of mobility. Applying strategies of I4.0 in mobility transportation businesses and construction as well as transportation systems could influence the development of Blue-Green mobility. In addition, digitalization is the other policy founded on technology used by European countries to develop sustainable and smart mobility. Applying digitalization to manage waste, make waste as a source of energy for transportation, creating smart transportation systems and so on are policies to make Blue-Green mobility. Ubiquitous services in transportation and all services including u-businesses, u-education, u-library and etc., are policies used in eastern countries like South Korea to gain Blue-Green mobility. Technology is the main indicator tool to develop U-services. Therefore, technology is a fundamental means in several ways to develop Blue-Green mobility as a smart and sustainable mobility system. Figure 7 presents the technology systems used in western and non-western countries to develop Blue-Green mobility technologies:

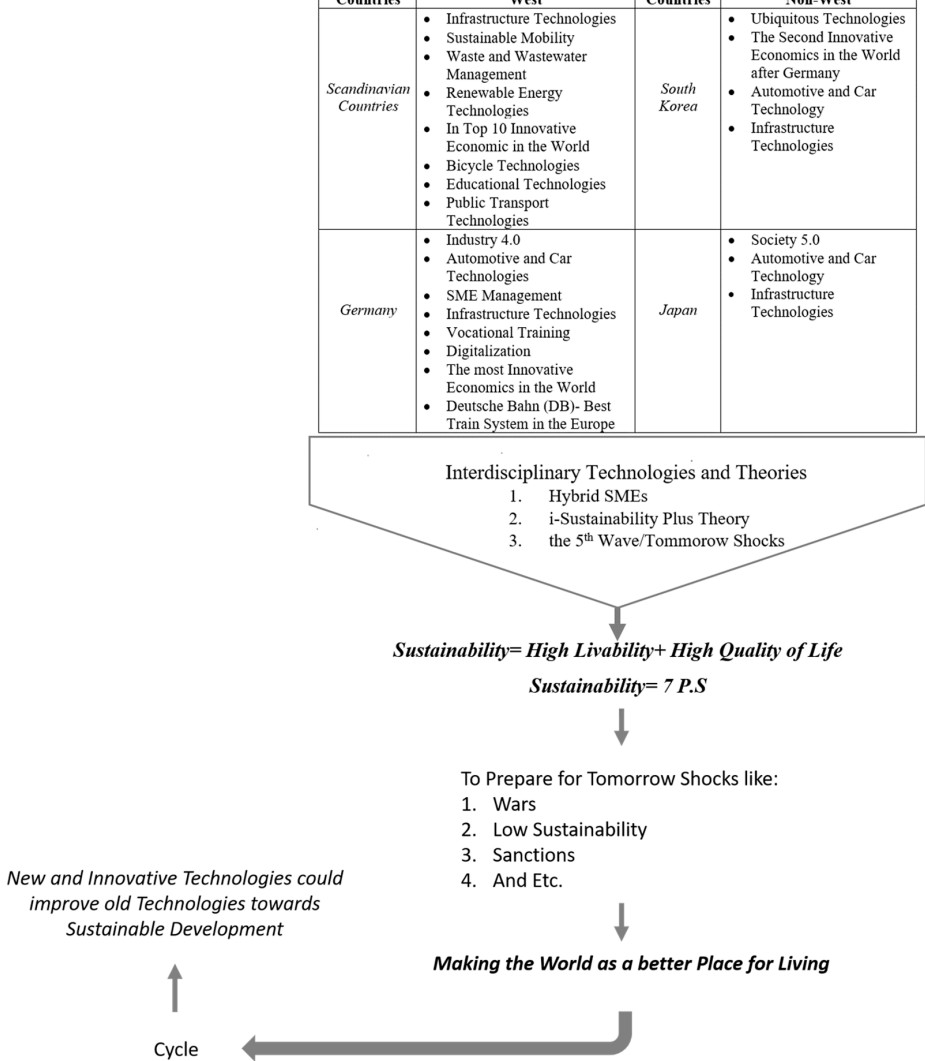

| Countries | West | Countries | Non-West |
|---|---|---|---|
| *Scandinavian Countries* | • Infrastructure Technologies<br>• Sustainable Mobility<br>• Waste and Wastewater Management<br>• Renewable Energy Technologies<br>• In Top 10 Innovative Economic in the World<br>• Bicycle Technologies<br>• Educational Technologies<br>• Public Transport Technologies | *South Korea* | • Ubiquitous Technologies<br>• The Second Innovative Economics in the World after Germany<br>• Automotive and Car Technology<br>• Infrastructure Technologies |
| *Germany* | • Industry 4.0<br>• Automotive and Car Technologies<br>• SME Management<br>• Infrastructure Technologies<br>• Vocational Training<br>• Digitalization<br>• The most Innovative Economics in the World<br>• Deutsche Bahn (DB)- Best Train System in the Europe | *Japan* | • Society 5.0<br>• Automotive and Car Technology<br>• Infrastructure Technologies |

Interdisciplinary Technologies and Theories
1. Hybrid SMEs
2. i-Sustainability Plus Theory
3. the 5th Wave/Tommorow Shocks

*Sustainability= High Livability+ High Quality of Life*

*Sustainability= 7 P.S*

To Prepare for Tomorrow Shocks like:
1. Wars
2. Low Sustainability
3. Sanctions
4. And Etc.

*New and Innovative Technologies could improve old Technologies towards Sustainable Development*

***Making the World as a better Place for Living***

Cycle

**Figure 7.** Western and Non-Western Blue-Green Mobility Technologies.

Fundamentally, Blue-Green mobility is based on Blue-Green transportation and sustainable mobility businesses. Therefore, Blue-Green mobility has two aspects, and technology plays roles in both aspects of mobility.

Blue-Green transportation is concerned with safe, efficient, sustainable, and smart transport services through intelligence, smart transportation, omnipresent services, ubiquitous technologies and environmentally friendly strategies. Such transportation could keep up with growing population and new demands of citizens, even to be aligned with developing sustainability. Blue-Green transportation like U-transport could create an environment founded on existing networks everywhere with consumer durable devices, dispersion of intelligence and information, being always accessible and smart services. It is able to make global networks that will be accessible and available anytime, anywhere with high quality services for anyone. In particular, this transport system could solve traditional transport system challenges such as: clean, affordable and sustainable propulsion, proper and sustainable infrastructure to gain smart personal mobility and logistics, secure and safety connected, cooperative and automated mobility and transportation and smart interaction among users and vehicles. Technology is a path to develop Blue-Green mobility. Technology is the main factor of vehicles working with clean energies, digital transportation services, the ubiquitous feature of being everywhere, anytime for anybody and so on, that are required for Blue-Green transportation. For instance, uber as an application to make transportation easier for anybody is founded on this improvement of technology.

As it was mentioned, sustainable mobility and transportation businesses are needed to gain Blue-Green mobility. Hybrid businesses which focus on environmentally friendly, sustainable water management and CSR strategy are needed for Blue-Green mobility. These businesses, through sustainable and Blue-Green management, risk management, CSR strategies based on social, economic, and environmental sustainability, could improve environmental sustainability. Technology is a tool to develop these strategies towards Blue-Green mobility. In addition, such hybrid organization could improve social and economic sustainability. All these solutions are based on technology.

In addition, sustainable road and urban construction would influence creating Blue-Green mobility. For instance, the increase of natural disasters like floods is related to environment and nature disruption by humans. Hybrid construction organization based on sustainable and intelligent management, CSR strategies, using clean energies, open space strategy, urban forest strategy, biodiversity strategy, waste technology, reducing nature disruption could impact on Blue-Green mobility. To attain these strategies, technology plays a significant role.

Fundamentally, technology is vital to create Blue-Green mobility and all solutions and strategies needed to make such mobility are based on technology. However, technology plays significant roles in creating Blue-Green mobility; it is required to train smart citizens who are aware of technology and services founded on technology. Smart citizens are ready to confront with future shocks and could struggle with them. Society 5.0 is a good example of modern society that is based on technology with smart citizens. Smart citizens are able to utilize smart and technology-based services and be aligned with modern services. These are vital to create a smart and intelligent society like society 5.0 that could improve economic and social sustainability as well as environmental sustainability. Such a society would improve quality of livability and life besides sustainability. Basically, smart citizens are needed to tackle future shocks related to rapid improvement of technology.

Technology is a doubled-edged sword as a fundamental tool of developing sustainability and even making future shocks. Education would train smart citizens towards smart society that will benefit from technology privileges.

In this research, *i*-Sustainability plus is declared by authors to develop Blue-Green mobility. It has three main components: Ubiquitous computing, Innovation and Sustainability. Figure 8 presents **_i_-Sustainability Plus Theory**:

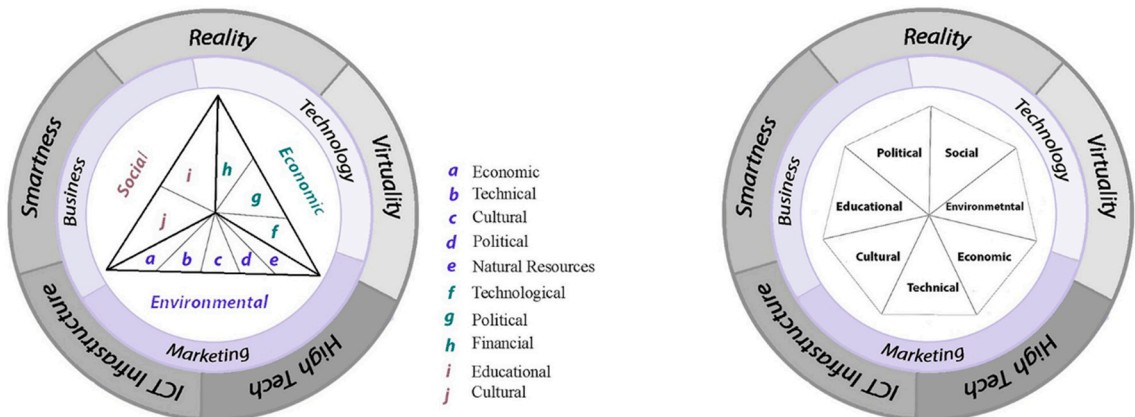

**Figure 8.** *i*-Sustainability Plus Theory [8].

As it was mentioned, technology plays a role in all three components of this theory. Ubiquitous is itself based on technology and its services could be developed through technology. In addition, sustainability strategies and innovation solutions could be made by technology. In the mobility section, IoE-Energy Management, vehicles working with clean energies, hybrid organization, sustainable constructions and other solutions are all achieved by technology. Furthermore, innovative solutions like *i*-Sustainability Plus is founded on technology. So, technology is a substantial tool to gain clean and inclusive mobility like Blue-Green mobility, U-transportation, smart transportation and so on.

*6.3. Blue-Green Mobility and Sustainability*

Blue-Green infrastructure not only improves sustainable development, but also supplies humans' needs such as security, economic development, education, housing, etc., perfectly. These needs are made into indicators required for livability and quality of life. Based on the research, livability is concerned with political stability, safety, healthcare, education, public services, transportation, recreation, housing, and environmental quality and quality of life is estimated through factors such as physical aspects like the quality of the built environment, its facilities, and social aspects such as the human ties in the community. These two parameters could be supplied by Blue-Green infrastructure.

Based on Prof. Dr. Doost Mohammadian's model, introduced in 2018 entitled: 5N.BG.7PS model, successful Blue-Green mobility technology development projects would include five networks:

1.  The political networks
2.  The technical networks
3.  The organizational networks
4.  The economic networks
5.  The interdisciplinary community networks

Figure 9 presents the relation among these five aspects, Blue-Green Sustainability, Quality of life and Livability (5N.BG.7PS model):

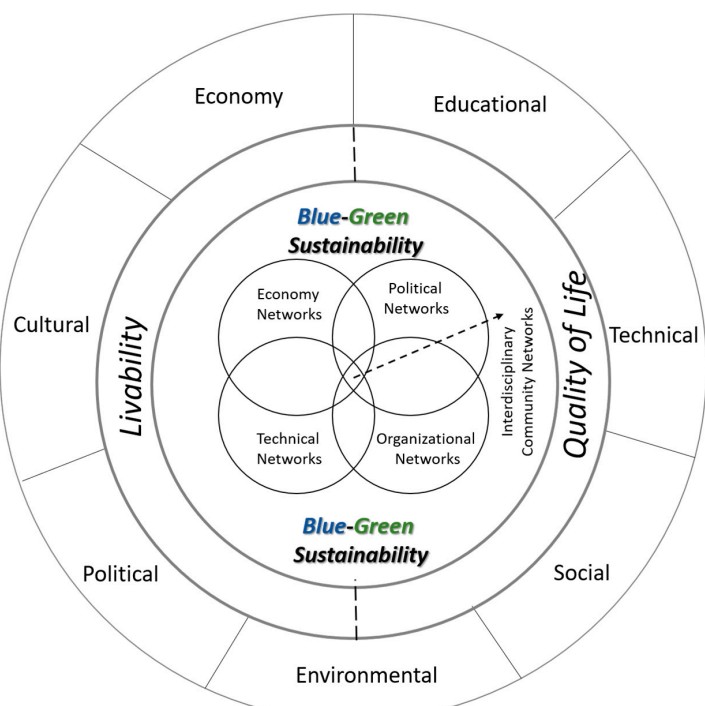

**Figure 9.** 5N.BG.7PS Model.

Generally, Blue-Green mobility can improve livability and quality of life through supplying humans' needs. Furthermore, mobility through comprehensive strategic urban planning and sustainability could be influenced by infrastructure. Mobility systems are related to sustainability in two aspects. First, Blue-Green, and inclusive mobility develop sustainability in order to improve quality of livability and life. It focuses on struggling with urbanization challenges and supplying humans' needs in order to enhance livability and quality of life. Blue-Green transportation like enhancing public transportation, vehicles worked with clean energies, etc., are able to reduce environmental challenges in order to improve livability. So, such mobility through reducing urbanization challenges would supply humans' needs and improve sustainable development in order to create livable and

sustainable urban areas. Second, there is a bilateral relation among Blue-Green mobility and the seven pillars of sustainability. It means that strategies concerned with sustainability are required to attain Blue-Green mobility, and such mobility could improve sustainability. Based on the authors' concept, sustainability has seven pillars instead of three traditional pillars. Economic sustainability, environmental sustainability, educational sustainability, cultural sustainability, social sustainability, technical sustainability, and political sustainability make seven pillars of sustainability. For instance, technical sustainability concerned with vehicles working with clean energy, educational sustainability as a tool to improve knowledge about sustainability and altering cultural norms, political sustainability concerned with global standards, etc., are needed to achieve Blue-Green mobility. In addition, such mobility could gain seven pillars of sustainability. Vehicles based on hybrid and clean energies, enhancing public transportation, promoting cultural norms and other policies of Blue-Green mobility could improve the seven pillars of sustainability. So, there is a bilateral relation between mobility and sustainability. Furthermore, mobility systems, through comprehensive strategic urban planning, can affect infrastructure. Ccomprehensive strategic urban planning should focus on services based on innovation, smartness, digitalization and ubiquitous to gain Blue-Green mobility. Uber, E-services like E-education, E-banking, E-shopping and other E-services, using innovative and sustainable risk management in order to reduce mobility risks, utilizing innovative mobility systems like strategies and policies that Maersk Danish mobility uses, etc., should be added in urban planning as tools required for improving Blue-Green mobility.

Figure 10 presents *i*-LUA Theory that is suggested by authors to present impacts of mobility systems on comprehensive strategic urban plann and sustainability.

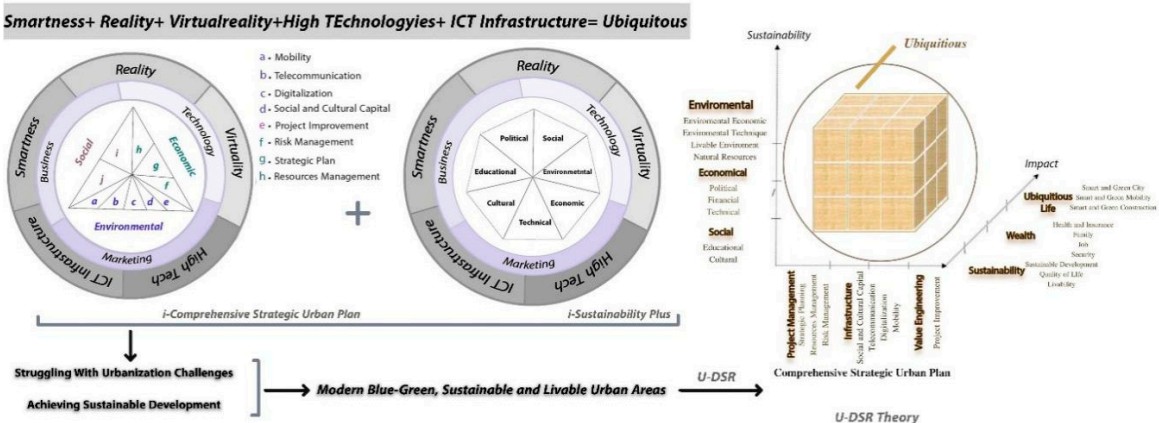

**Figure 10.** *i*-LUA Theory on Mobility.

### 6.4. Modern Livable Urban Areas

Blue-Green mobility could create Blue-Green, smart, and sustainable infrastructure that is needed to design modern urban areas. Sustainable and smart urban areas require sustainable and smart mobility to be kept up with rapidly changing human demands and needs. Mobility as a part of infrastructure could influence the sustainability and livability of urban settings. Fundamentally, Sustainable Blue-Green mobility could create modern, sustainable, and livable urban areas through struggling with global challenges and achieving sustainable development. In particular, mobility as a component of a comprehensive strategic urban plan via other components including project management, value engineering and infrastructure, would create modern urban areas concerned with sustainability, wealth and ubiquitous life. These dimensions of modern areas are able to deal with growing urbanization, its challenges, risk mitigation and improve economic as well as social systems, create new opportunities for jobs, education, wealth, enhance smart life, etc., which are concerned with supplying humans' needs. Figure 11 presents Comprehensive Sustainable Mobility Theory which is suggested by authors to present impacts of mobility system on comprehensive strategic urban plan and sustainability.

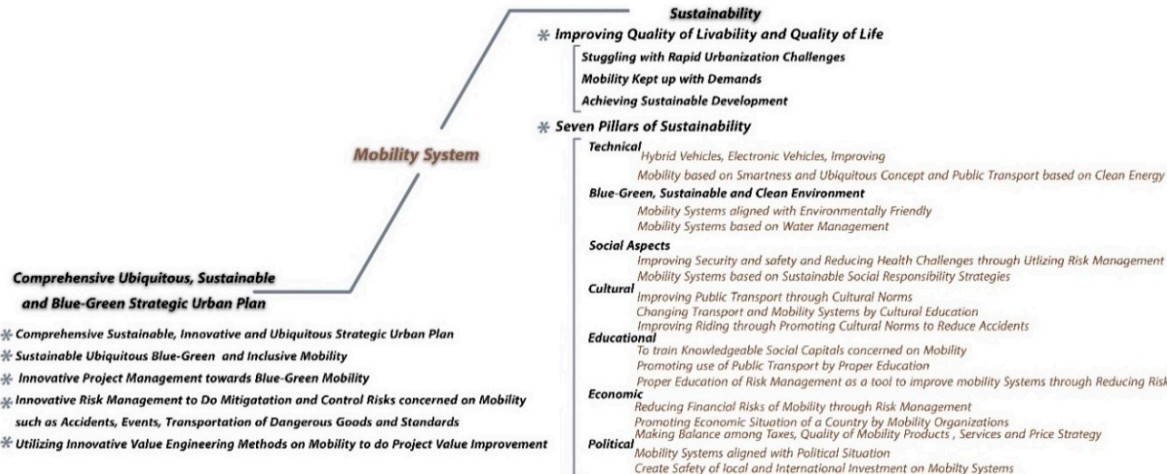

**Figure 11.** Comprehensive Sustainable Mobility Theory.

In this study, *i*-sustainability plus theory and *i*-comprehensive strategic urban plan are declared as solutions to design livable and sustainable areas based on U-DSR theory. *i*-Sustainability Plus is based on seven pillars of sustainability (7PS model), innovation management and ubiquitous ideas. This theory could be used in transportation systems to create Blue-Green mobility. In addition, comprehensive strategic urban planning based on innovation management and ubiquitous services is recommended as the other tool to create modern urban areas. As mobility is a component of urban planning, designing proper strategic urban plans is vital for modern urban settings. Therefore, *i*-comprehensive strategic urban planning focuses on urban planning including infrastructure, value engineering and project management, innovation management and ubiquitous services is realized as a tool for modern urban areas. These two theories could design modern livable urban areas with high quality of life, sustainability, wealth, and ubiquitous life named U-DSR Plus theory. This theory was realized by authors to create modern urban areas with high quality of livability and life. Therefore, U-DSR Plus is a path to create Blue-Green Cities towards sustainable and livable world. Figure 12 presents U-DSR Plus Theory.

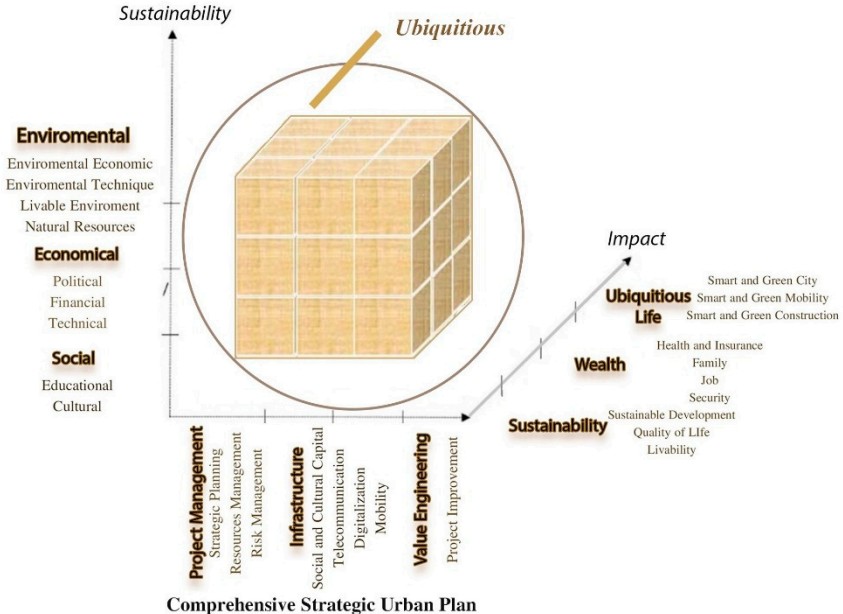

**Figure 12.** U-DSR Plus Theory.

Generally, technology is a component of these two theories. Technology exists as not only a ubiquitous part of these theories, but also in innovation and sustainability. In other words, technology is a tool to develop *i*-Sustainability Plus and *i*-Comprehensive Plus theories.

In particular, urban cities based on U-DSR theory are modern sustainable areas with sustainability, wealth and ubiquitous life that are able to create high quality of livability and life through their features. In addition, such areas are based on sustainable development, which could make the world as better place for living.

*6.5. Technologies as Readiness to Make the World a Better Place for Living*

Clean and inclusive mobility technologies based on Blue-Green and sustainable infrastructure, low emission greenhouse gases, ubiquitous, smartness and digitalization are realized as one of the keys that could make the world more livable and sustainable. Blue-Green sustainable inclusive mobility technologies, in addition to cultural synergies as readiness for facing tomorrow's shocks, make the world safer by using the 5th wave theory.

The 5th wave/tomorrow age theory could be a technique that is required to get ready for the edge of tomorrow, today's challenges, and tomorrow's crises in the field of transportation to create a sustainable world for present and future businesses. Figure 13 presents modern technologies for making the world a better place for living.

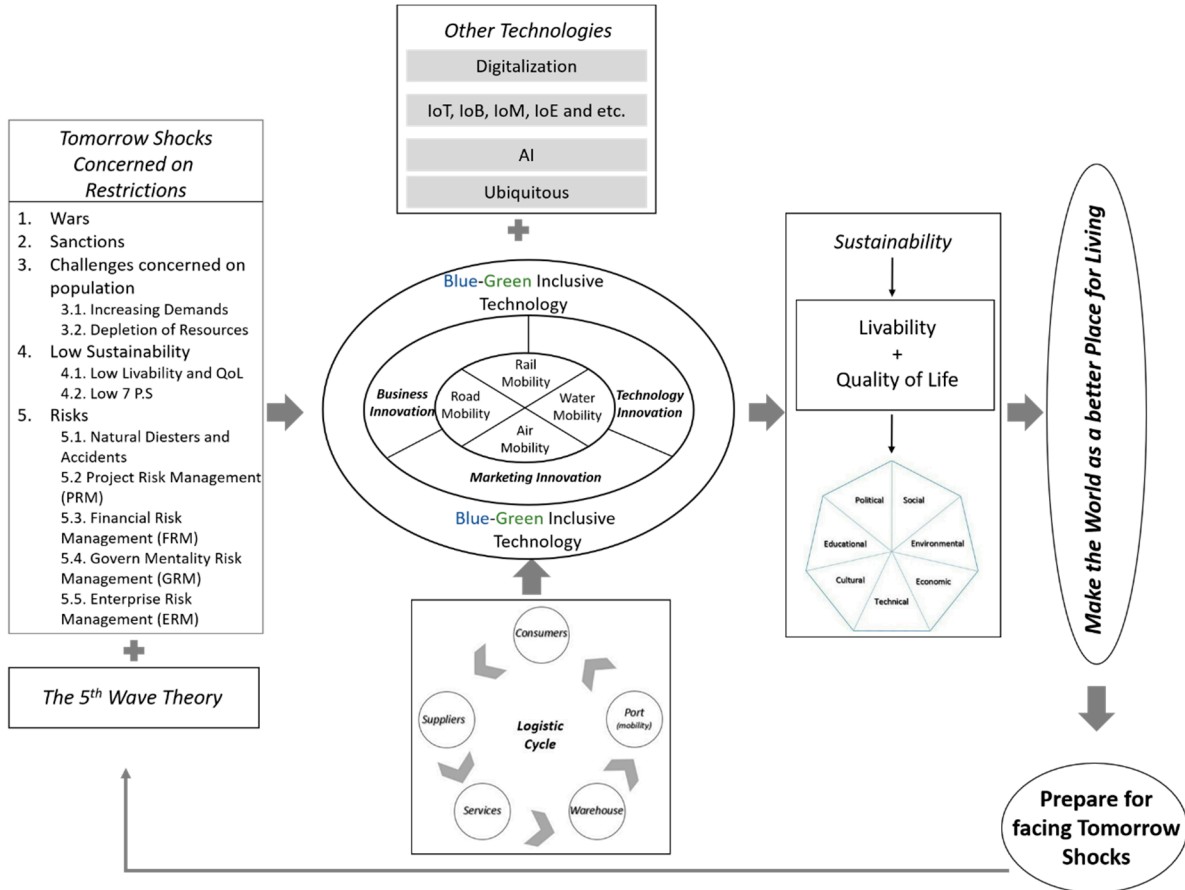

**Figure 13.** Technologies for making the World a Better Place for Living.

## 7. Conclusions

Nowadays, globalization and global challenges create challenges, and the world and urban areas need to be changed in order to deal with these challenges, like environmental challenges; climate change, economic problems, social instability, health challenges and insecurity, which threaten human

beings and the planet, to maintain humanity and the world for the future. New concepts of urban areas like ubiquitous city, digital city and smart city are introduced as paths to tackle these challenges. All these ideas are based on technology. So, technology is a tool to develop concepts, solutions, and strategies to reduce global challenges. However, technology can cause future shocks that can confuse humans; it is vital that they are applied in a way in which humans are aware of them and are able confront these future shocks, even to develop solutions needed for modern urban areas [8,19,41,66].

Fundamentally, mobility as a part of urban infrastructure could influence the design of modern sustainable and livable urban areas. Therefore, it is vital that governments, planners, architects, construction workers, organizations, and policy-makers work together to create clean and inclusive mobility in order to deal with urbanization challenges and achieve sustainable development to improve quality of life and livability.

Blue-Green mobility based on environmentally friendly and water management policies could create the clean and inclusive mobility required for designing modern urban areas like ubiquitous cities, and Blue-Green cities. Technology would play significant and fundamental roles in developing Blue-Green mobility. Vehicles working with clean energies, sustainable wastewater management, sustainable road construction, sustainable and hybrid transportation organization all are technology-based.

According to research, the below points are concluded:

1.  Creating modern sustainable and livable urban settings based on sustainable development are the best keys towards creating better world for human life.
2.  Mobility as a component of urban infrastructure has an important role in achieving such urban areas. Mobility could play a role in making sustainable and Blue-Green infrastructure that is concerned with the environment, water management, Blue-Green Strategy and the eeconomy.
3.  Clean and inclusive mobility is a mobility system aligned with sustainability, smartness, Blue-Green infrastructure and urbanization demands that would be achieved through sustainability, digitalization, smart planning, ubiquitous concept, proper education, knowledgeable, expert and vocational social capital, innovation and innovative procedure.
4.  Blue-Green mobility could influence other pillars of high sustainability in order to achieve sustainability. Improving sustainability impacts on quality of life and livability. Therefore, Blue-Green mobility would make the world as a better place for living through improving livability and quality of life.
5.  Technology is a tool to advance strategies, solutions and polices needed for Blue-Green mobility. In other words, technology needs to create different kinds of clean and inclusive mobility, such as Blue-Green mobility.
6.  Although technology plays a fundamental role in creating clean and inclusive mobility, it is a threat to make future shocks. Smart citizens are needed to be ready for future shocks and confronting them. Fundamentally, training smart citizens who are aware of the importance and usage of technology could influence future shocks as well as applying technologies in different services.
7.  Besides clean and inclusive mobility, the other components of comprehensive sustainable strategic urban plan are required to attain such areas. These indicators have an impact on creating a better world for living through modern sustainable and livable urban areas.

**Author Contributions:** Writing—review & editing, H.D.M. and F.R. All authors have read and agreed to the published version of the manuscript.

**Funding:** This research received no external funding.

**Conflicts of Interest:** The authors declare no conflict of interest.

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
