# Peer review of "Blue-Green Smart Mobility Technologies as Readiness for Facing Tomorrow’s Urban Shock toward the World as a Better Place for Living (Case Studies: Songdo and Copenhagen)"

_technologies, doi:10.3390/technologies8030039_

Round 1

Reviewer 1 Report

The authors present an interesting analysis of use-cases and compare different approaches of "smart cities". The study is very thorough and provides a good overview. There is some repetition from the author's previous work, but the current article has enough novelty. I would only recommend to adjust the title, it is not a good reflection of the work presented.

Author Response

Dear Reviewer,

Thanks for your comments and suggestions.

According to your comments, it was tried to reduce repetition of previous work and make alternatives for repetition like Songdo instead of Seoul as case study.

In addition, background section was reduced and concluded.

Title is changed

Reviewer 2 Report

The paper Blue-Green mobility technologies as readiness for facing today challenges and tomorrow shocks - to make the world a better place for living (Case Studies Seoul and Copenhagen) discusses regarding Blue-Green sustainable inclusive mobility technologies in addition cultural synergies as readiness for facing tomorrow shocks to make the world as a better place for living by using the 5th wave theory.

The paper does not bring significant scientific contributions but is rather like a literature review and more of a course material and not a research.

Parts of this paper (more exactly 13% of it) was previously published in Inventions (https://www.mdpi.com/2411-5134/5/2/14) as stated by the plagiarism platform TurninIn. Though the percentage may look small, there are big blocks of text that can be found in both articles. For instance, the Seoul case study can be entirely found in both papers (pages 15-16) also the text in page 17.

Issues:

-too many keywords(13), the Journal recommend 3 to 1

-line 104 replace "The bellowed figures show" with "In the two charts included in figure 1 are shown"

-line 173 replace "The below figure" with "Figure 2". the same for lines 261, 276, etc. 

-line 201 replace figure 2 with 3 and reference this figure from text!

-in line 286 reference figure 6 from text; same for figure in line 480 and 901

-in line 362 replace "below figure" with "figure 7"; same for line 433, 487,560, 621, 701, 727,760,813,862

-in line 438 eliminate the question mark(?) in the end and replace it with :

- the methodology fails to prove that is reliable in selecting with high accuracy the papers: what databases were used (ISI, Ebsco, etc ..)?, what period was included in the analysis?, what keywords were used in the paper selection and how many results were obtained? How were selected the final list pf papers? etc..

-line 489 this sentence is very vague : Based on literature reviews, case study, surveys and analyses, some results were obtained . Please add the numbers of all the above types papers.

-elaborate more on the results.

Author Response

Dear Reviwer,

Thanks for your valuable comments, motivation and suggestions

Parts of this paper (more exactly 13% of it) was previously published in Inventions (https://www.mdpi.com/2411-5134/5/2/14) as stated by the plagiarism platform TurninIn. Though the percentage may look small, there are big blocks of text that can be found in both articles. For instance, the Seoul case study can be entirely found in both papers (pages 15-16) also the text in page 17

In revised version, we have reduced repetition of previous work e.g. We introduced Songdo instead of Seoul as case study in South Korea

Issues:

-too many keywords (13), the Journal recommend 3 to 1: 3. keywords reduction has been done

-line 104 replace "The bellowed figures show" with "In the two charts included in figure 1 are shown": has been done

-line 173 replace "The below figure" with "Figure 2". the same for lines 261, 276, etc. we have deleted Figure 2

-line 201 replace figure 2 with 3 and reference this figure from text! has been done

-in line 286 reference figure 6 from text; same for figure in line 480 and 901 has been done

-in line 362 replace "below figure" with "figure 7"; same for line 433, 487,560, 621, 701, 727,760,813,862 has been done

-in line 438 eliminate the question mark(?) in the end and replace it with:

- the methodology fails to prove that is reliable in selecting with high accuracy the papers: what databases were used (ISI, Ebsco, etc ..)?, what period was included in the analysis?, what keywords were used in the paper selection and how many results were obtained? How were selected the final list pf papers? etc..

We describe the research methodology with the detailed information

-line 489 this sentence is very vague : Based on literature reviews, case study, surveys and analyses, some results were obtained . Please add the numbers of all the above types papers. has been done

-elaborate more on the results. has been done

The other points are done based on your comments, In addition, all changes were written in red color

Best wishes and stay healthy

Reviewer 3 Report

This paper focuses an interesting topic. Literature is rich and updated. Anyway methodology should be rewritten enphasizing also by means of literature the importance of each research question.

Results are interesting but I suggest to emphasized also research limits.

Author Response

Dear Reviewer,

This paper focuses an interesting topic. Literature is rich and updated. Anyway methodology should be rewritten enphasizing also by means of literature the importance of each research question.

Thanks for your comments, motivations, and suggestions. Methodology were improved with more detailed information by means of literature the importance of each research question and abstract was modified.

Results are interesting but I suggest to emphasized also research limits.

In revised version, it was tried to reduce and conclude literature review and focus more on results, discussion and analysis.

Best wishes and stay healthy

Round 2

Reviewer 2 Report

The article Blue-Green Smart Mobility Technologies as Readiness for Facing Tomorrow’s Urbans’ Shocks to Make the World a Better Place for Living (Case Studies: Songdo and Copenhagen) discusses a new concept of mobility technology in addition cultural synergies as readiness for facing tomorrow’s crises to make the world as a better place for living by using the 5th wave theory. Though interesting, the paper has many typos and inconsistencies with the declared research paths. Some typos:

Line 247 please change Figure 5 with Figure 2;

Line 257: Figure 3 is not referred from text, please change;

Line 373: remove the question mark in the end;

Figures 6,7,8 and 9 are not referred from text, please change;

Line 579 and 586: Tables' titles is located above of the table not below;

Line 580: please define M2 and M1;

Starting with Figure 10 (line 690) the authors are not consistent with the number of Figures when they refer it from text:

-line 687 replace Figure 11 with Figure 10;

-line 713 replace Figure 12 with Figure 11;

-line 745 replace Figure 13 with Figure 12;

-line 795 replace Figure 14 with Figure 13;

-line 889 replace Figure 17 with Figure 16;

Line 847 Figure 14 is not referred from text; please do so;

Line 895, in the end, please change "aeras" with areas;

Line 941 please change the font.

In the abstract (which is too long in my opinion) the authors should also write about the two cities study.

About the methodology, the same issue as in the previous review:  the methodology fails to prove that is reliable in selecting with high accuracy the papers: what databases were used (ISI, Ebsco, etc ..)?, what period was included in the analysis?, what keywords were used in the paper selection and how many results were obtained? How were selected the final list pf papers? etc.

The antiplagiarism check returns a 22% plagiarism rate(please check the attached file) and 14% is from mdpi.com platform. I suggest to scan it also and try to reduce this rate.

Author Response

Dear Reviewer:

Thanks for your valuable comments and guidance. New version of the article was revised through your comments. All changes were illustrated in red color.

The main ones are:

  • Line 247 (in new version line 242) was revised.
  • Line 257 (in new version line 249) was revised and figure 3 was declared and referred in the context.
  • Line 373(in new version line 369) was revised.
  • Figure 6,7 and 9 were referred from text.
  • Line 579 (in new version line 574) was modified.
  • This methodology analysis was removed.
  • Line 687 (in new version line 681) was revised.
  • Line 713 (in new version line 707) was revised.
  • Line 745 (in new version line740) was revised.
  • Line 795 (in new version line 790) was revised.
  • Line 889 (in new version line 883) was revised.
  • Line 895 (in new version line 889) was modified.
  • Line 941 (in new version line 935), the font was modified.

  • Abstract was revised based on Your guidance and concluded.

  • It was tried to modify methodology and result sections according to Your comment and guidance. Details about methodology were added to make it more specific.

A new questionnaire between 30 experts in sustainability field was done. New questions focus on relation between sustainable and smart mobility and quality of life indicators. It presents that smart and sustainable mobility on which QoL indicators influence more.

According to time, this questionnaire was done among only 30 persons and just five main indicators are analyzed.

  • Eventually, it was tried to reduce the similarity rate that you have mentioned through scanning.

Best wishes

Hamid Doost Mohammadian

Round 3

Reviewer 2 Report

The authors have dealt with all my suggestions. Also, they reduced the plagiarism rate from 22% to 17%.

Therefore, I consider the new version of the manuscript is suitable for being published with Technologies Journal.

This manuscript is a resubmission of an earlier submission. The following is a list of the peer review reports and author responses from that submission.

Round 1

Reviewer 1 Report

The authors seem to have put too many concepts that they don’t develop, and sounds like buzz words, such as Information Technology, Internet of Things, Internet of Business, Internet of 72 Manufacturing and Internet of Energy. It also reflect in the fact that the paper has 12 keywords. In the background session something similar occurs: the authors mention Ubiquitous city, smart city, E-services, Ubiquitous and smart services, IoE-143 energy management, Industry 4.0 and so on are based on modern technologies like ICT, IoT and IoE; then only define a few of them, but define them extensively, beyond the need to make them instrumental to their analysis.

The introduction mentions several times Blue-Green mobility, but it does not present any clear definition of what this concept actually is. Also, authors name Köhler, but don’t give any reference.

In the background, the authors spend too many paragraphs explaining well-known ideas about mobility, and include common-sense phrases that don’t seem relevant in a scientific paper, such as: “high quality of livability and life, health and prosperity with social justice, being environmentally friendly and preserving the earth's capacity to support future life are the main aims of the sustainable development”.

The paper present 8 research questions; thus, I was looking to see answers for each of them. They seem to be there, but not in a organized way. The authors say they had a survey, so I was expecting to see the interview protocol, the profile of the interviewees, and a systematic analysis of the surveys.

By the end of the paper, authors summarize Blue-Green mobility, where Green represents policies like sustainability and environmentally friendly, and Blue mobility related to water management. At this point, I got even more confused: what about cities that don’t have waterways? Or, if the blue part of the concept relates to any sort of water management, why the concept is linked to mobility?

The paper has good ideas; however, at this stage it is simultaneously excessively long and extremely generic. In order to improve the paper the authors could: a) summarize the concept from the get-go; b) summarized the conceptual background; c) design and present a clear empirical test of their ideas. Otherwise, the contribution is minimal for such long text.

Reviewer 2 Report

The paper is long and unfocused.  It start with 13pages of description of general concepts (which is redundant with other published papers), methodology takes less than a page (which clearly worrisome).  The paper should be completely refocused on what is new and unexpected in what authors see and probably case studies should be driving it.  On methodological side, the whole concept of resilience is missing.  Since authors talk about shocks and stressors, resilience should be discussed.  A good starting point may be Linkov, I., & Trump, B. D. (2019). The Science and Practice of Resilience. Springer, Amsterdam; and Bostick, T.P., et al (2018).  Resilience Science, Policy and Investment for Civil Infrastructure. Reliability Engineering & System Safety 175:19-23.